# Context-Aware World Models for Task-Agnostic Control

**Busra Tugce Gurbuz**[1,3,4,5*]     **Hafez Ghaemi**[2,3,4*]
**Christopher C. Pack**[1,5]     **Shahab Bakhtiari**[2,3]     **Eilif B. Muller**[2,3,4]

[1]McGill University     [2]Université de Montréal
[3]Mila - Quebec AI Institute
[4]Centre de Recherche Azrieli du CHU Sainte-Justine  [5]Montréal Neurological Institute

## Abstract

Building accurate and generalizable world models requires aligning an agent's inductive biases with the structure of the environments it inhabits. Realistic environments are inherently multi-task, containing both shared global dynamics and context-specific variations that differ across tasks. We argue that a world model should reflect this structure by efficiently separating context-specific dynamics while integrating shared regularities that support generalization. As a first step towards this, we introduce Context-Aware World Models (CaWM), which align the agent's world model with the multi-task structure of the environment. In contrast to existing model-based approaches that assume access to ground-truth task labels, CaWM learns to infer latent task contexts directly from its interactions with the environment via a self-supervised objective, and uses these inferred contexts to modulate the world-model representations, enabling task-agnostic control. To benchmark CaWM, we present Multi-FoE, a multi-task visual foraging environment with egocentric partial observations and boundary-free task switching. Empirically, CaWM achieves higher performance and success rate compared to context-free baselines, and approaches the performance of an oracle with ground-truth task labels.

## 1 Introduction

Building world models that generalize across tasks remains a central challenge in model-based control and reinforcement learning. Most existing approaches implicitly assume that the agent operates in a single-task environment or that the identity of the underlying task is externally provided through ground-truth labels or oracle embeddings [Kanitscheider et al., 2021, Hansen et al., 2023, Hafner et al., 2025]. Yet realistic environments rarely provide such supervision. Instead, they exhibit latent structure: stable global dynamics shared across tasks, intertwined with context-dependent variations that must be inferred directly from interaction [Wang et al., 2017, Duan et al., 2016, Rakelly et al., 2019]. World models that conflates these sources of variation risk to either overfit to individual patterns within individual tasks or collapse into an averaged dynamics model that fails to adapt to context-specific changes [Humplik et al., 2019, Yu et al., 2020].

We argue that world models that are effective in task-agnostic control must explicitly model latent context. In particular, the agent should autonomously infer the task context from its own interaction trajectory and use this inferred context to modulate both its internal state dynamics and its control

---

*Equal contribution.
Correspondence: tugce.gurbuz@mail.mcgill.ca, hafez.ghaemi@umontreal.ca

39th Conference on Neural Information Processing Systems (NeurIPS 2025) Workshop: Unifying Representations in Neural Models (UniReps).

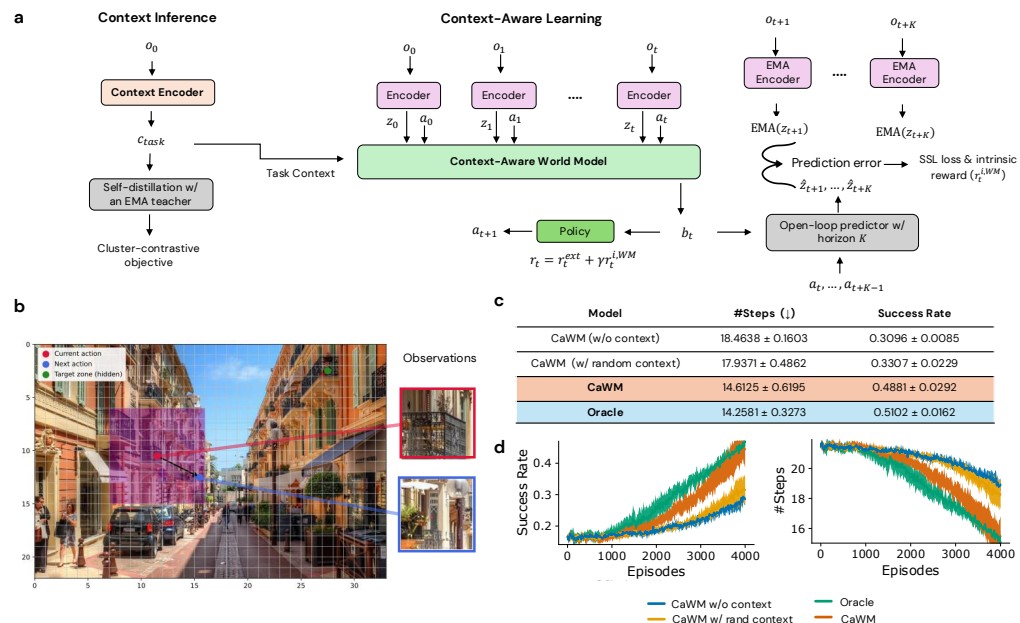

**Figure 1: (a)** *Context-Aware World Model (CaWM)*; context inference module infers a task context via self-supervised learning and guides context-aware learning trained with a self-supervised prediction loss, an intrinsic reward, and an external task reward (if available). **(b)** *Illustration of an example MultiFoE task.* The red frame depicts the foveated observation centered on the agent's current gaze location (red dot). The blue frame shows the subsequent observation following the selected next action (blue dot). The purple square outlines the egocentric action set available to the agent. The green dot indicates the hidden target zone the agent must discover. **(c-d)** Comparative results for CaWM and baseline models.

policy. This mirrors how biological systems flexibly adjust predictions and strategies when environmental conditions shift [Friston, 2010]. Building on this motivation, we propose Context-Aware World Models (CaWM), that infer task contexts through self-supervised learning on its interaction data to enable context-aware world modeling and task-agnostic control.

To evaluate CaWM, we introduce Multi-FoE, a visual foraging benchmark environment consisting of 20 distinct visual foraging tasks. Empirically, CaWM consistently surpasses context-free baseline and approaches the performance of an oracle with access to ground-truth task labels, highlighting its ability to achieve reliable task-agnostic control.

## 2 Method

Our model architecture (Figure 1.a) consists of two components: a *context inference module* and an *context-aware world model*.

**Context Inference Module:** At the beginning of each episode, the context encoder receives the initial observation ($o_0$) and infers a latent embedding of the task context. This embedding is used as a top-down prior that conditions the world model. Note that many task structures cannot be separated from a single observation alone, in which case, information about dynamics, e.g., $(o_t, a_t, r_t, o_{t+1})$ (potentially sampled from a replay buffer), should be provided to the context inference module. The context inference module can be trained via a joint-embedding self-supervised learning (SSL) algorithm [Chen et al., 2020, Grill et al., Bardes et al., 2022, Caron et al., 2021]. Such SSL algorithms require pairs of data points with invariant features (i.e., "positive" pairs). In our setup, the pairs are $(o_{0,i}, o_{0,j})$ where $i$ and $j$ denote two different episode trajectories. To obtain these $(o_{0,i}, o_{0,j})$ pairs, we leverage the parallel rollouts naturally generated by proximal-policy optimization (PPO) [Schulman et al., 2017], the RL algorithm we employ for learning a policy.

Specifically, we use DINO [Caron et al., 2021] as our SSL algorithm, which projects the context encoder outputs to cluster prototypes and minimizes a self-distillation *cluster-contrastive* objective

[Caron et al., 2021] to enforce invariances to the pair of representations corresponding to the same task:

$$\mathcal{L}_{\text{CC}} = \sum_{t=1}^{T} \sum_{\substack{m,n=1 \\ m \neq n}}^{M} H\big(P_{\text{teacher}}(c_{\text{task},t}^{m}),\ P_{\text{student}}(c_{\text{task},t}^{n})\big), \tag{1}$$

where $m$ and $n$ correspond to indices of two episode trajectories from a shared task, $P_{\text{student}}$ and $P_{\text{teacher}}$ are probability assignments over a set of learned cluster prototypes, and $H(\cdot, \cdot)$ denotes cross-entropy.

**Context-Aware World Model:** This module consists of a recurrent predictive world model (WM) based on BYOL-Explore [Guo et al., 2022] conditioned on inferred context $c_{task}$. At each step $t$, it updates its belief state using current latent observations $z_t$, actions $a_t$, and the task context $c_{task}$:

$$b_t = h_\theta(b_{t-1},\ a_t,\ z_t,\ c_{task}),$$

which is passed to the task policy. From $b_t$, the WM also performs *open-loop* rollouts of latent dynamics of length $K$ using its context-conditioned dynamics model, with predictions ($\hat{z}_t$) trained via a BYOL-style SSL loss against encoder targets (EMA($z_{t+k}$)):

$$L_{\text{SSL},t+1} = \frac{1}{K} \sum_{k=1}^{K} \left\| \frac{\hat{z}_{t+k}}{\|\hat{z}_{t+k}\|_2} - \text{sg}\left( \frac{\text{EMA}(z_{t+k})}{\|\text{EMA}(z_{t+k})\|_2} \right) \right\|_2^2. \tag{2}$$

Following BYOL-Explore [Guo et al., 2022], we define the world model uncertainty at step $t$ as the sum of open-loop prediction errors for all predictions that include the transition $(o_t, a_t, o_{t+1})$:

$$\ell_t = \sum_{p+q=t+1} \mathcal{L}_{SSL,t+1}(p, q),$$

which accumulates all the losses corresponding to the world-model uncertainties relative to the observation $o_{t+1}$. The intrinsic reward is proportional to $\ell_t$ and we follow the reward normalization scheme of Guo et al. [2022] to normalize it.

We use PPO to train the task policy and maximize the combined reward:

$$r_t = r_t^{\text{ext}} + \gamma\, r_t^{i,\text{WM}}, \tag{3}$$

where $r_t^{\text{ext}}$ is the external task reward (when available) and $r_t^{i,\text{WM}}$ is the normalized intrinsics reward.

Additional implementation details of the architecture and its components are provided in Appendix A.

## 3 Experiments

### 3.1 Multi-FoE Benchmark

To evaluate CaWM in a multi-task control setting, we desing *Multi-task Foveated Observation Environment (Multi-FoE)*, which is a multi-task visual foraging setup inspired by primate studies [Csorba et al., 2022]. The environment consists of 20 distinct task contexts (see Appendix B for all task contexts and corresponding target zones), each defined by a $1200 \times 800$ pixels natural scene discretized into a $22 \times 33$ spatial grid (an example task setup is shown in Figure 1.b). Within each scene, a hidden target zone is placed at a grid location. At every time step, the agent receives an $84 \times 84$ pixels foveated image centered on its current gaze location (red frame) and selects its next fixation (blue frame) from a $9 \times 9$ grid surrounding its center of gaze (purple square). The chosen saccadic action ( referred to as the action) corresponds to the displacement between consecutive gaze centers, enabling the agent to explore the scene through sequential, gaze-contingent observations that approximate the dynamics of active visual foraging in primates.

This setup also imposes a strict interaction budget per episode (25 steps, corresponding to $\sim 5$ seconds of visual search in humans [Najemnik and Geisler, 2008]). The limited horizon requires

the agent to actively select informative view points, and learn a control policy efficiently. Episodes terminate when the hidden target is located or the interaction budget is exhausted. Importantly, in this benchmark, task contexts alternate unpredictably between episodes without boundary cues, eliminating explicit supervision and requiring on-the-fly context inference for effective adaptation. Together, Multi-FoE brings core principles of embodied active vision into a controlled multi-task environment and offers a benchmark for evaluating agents under biologically grounded, realistic constraints. Complete details are provided in Appendix B.

### 3.2 Baselines and Evaluation Protocol

To isolate the contribution of our context inference mechanism, we compare CaWM against three baselines (Figure 1.c): (1) BYOL-Explore, which serves as a context-independent WM baseline, (2) an oracle variant, where ground-truth task labels are embedded as context, (3) CaWM variant that replaces the context embeddings with random Gaussian noise. Each seed was trained for $4,000$ episodes, with each episode capped at 25 saccadic steps, consistent with the baseline setting. At the beginning of each episode, the gaze was initialized at a random location on the spatial grid. An episode is terminated either when the agent saccades through the hidden target zone or when the maximum number of saccades is reached. Performance is quantified using two behavioral metrics: the average number of steps required to locate the target zone (efficiency) and the success rate (proportion of episodes in which the target was found). To calculate these metrics, we evaluated the model on all 20 task contexts every 10 episodes and also at the end of training. We report our results over six random seeds.

### 3.3 Results

Figure 1.d shows the evaluation performance throughout learning and Figure 1.c provides the evaluation results after $4,000$ learning episodes. Across all 20 task contexts, CaWM consistently outperforms the two baselines, achieving higher efficiency and success rate in locating hidden target zones. Notably, CaWM's performance matches that of the oracle model, demonstrating that it can infer task context solely from its own experience and without any supervision. Taken together, tthese findings show that CaWM offers a principled approach to context-aware world modeling, enabling agents to infer latent task structure from raw interaction and adjust their internal dynamics to support generalizable multi-task control.

## 4 Conclusion

We introduced Context-Aware World Models (CaWM), a framework for task-agnostic control that learns to infer latent task context directly from interaction and uses this context to modulate its latent dynamics for multi-task control. Our model departs from conventional world-model pipelines by explicitly decomposing environment structure into shared dynamics and latent task-specific variations, learned entirely through self-supervision. To evaluate this capability, we proposed Multi-FoE, a biologically inspired active-vision benchmark with boundary-free task switching, strict interaction limits and ego-centric partial observations. Across all tasks, CaWM consistently outperforms context-free world models and approaches oracle performance, demonstrating that reliable context inference is achievable without external supervision. Looking ahead, promising future directions include equipping CaWM with latent-planning and task-transfer capabilities, as well as scaling it to larger, more diverse multi-task benchmarks to evaluate its adaptability and robustness at greater scale.

## 5 Acknowledgments

This project was supported by funding from the Natural Sciences and Engineering Research Council of Canada (NSERC; Discovery Grants RGPIN-2022-05033 to E.B.M., RGPIN-2023-03875 to S.B., and RGPIN-2023-03853 to C.C.P.), the Canada CIFAR AI Chairs Program and Google (to E.B.M.), the Canada Excellence Research Chairs (CERC) Program, Mila – Quebec AI Institute, the Institute for Data Valorization (IVADO), the CHU Sainte-Justine Research Centre, the Fonds de Recherche du Québec – Santé (FRQS), and a Canada Foundation for Innovation (CFI) John R. Evans Leaders Fund grant to E.B.M. This research was also supported in part by the UNIQUE Centre and the Digital Research Alliance of Canada (DRAC). B.T.G. is supported by the Fonds de Recherche du Québec – Nature et Technologies (FRQNT).

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

# A    CaWM Implementation Details

For DINO, we use a ResNet-18 encoder pre-trained on ImageNet-100, and an MLP projector with 512-1024-1024 dimensions and ReLU activations and batch normalization.

For BYOL-Explore, we use a non-pretrained ResNet-18, a learnable affine action embedding of 128-d, and an LSTM with a hidden size of 512-d. For policy head, we use PPO [Schulman et al., 2017] with linear actor and critic heads that receive LSTM hidden states as inputs. For PPO training, we use 5 PPO training epochs, $\gamma_{PPO} = 0.99$, clip ratio of 0.2, value function coefficient of 0.5, target KL divergence of 0.015, $\lambda_{GAE} = 0.95$, and entropy coefficient of 0.01. The sequence length is set to 16 with a batch size of 128. For training, we use AdamW optimizer with a learning rate of $3e - 4$ and a weight decay of 0.0001 for all components.

# B    Multi-FoE Benchmark Details

## B.1    Benchmark Design and Biological Motivation

Multi-FoE combines concepts from embodied active vision with the core constraints of real-world learning. Its design is directly inspired by primate studies of visual foraging [Chukoskie et al., 2013, Csorba et al., 2022], and reflects four key principles fundamental to natural intelligence — active perception, context inference, goal inference, and sample-efficient learning — each of which introduces specific challenges that current models struggle with. Below, we outline these principles and their neuroscientific grounding to highlight the biological plausibility and task realism that Multi-FoE brings to multi-task agent evaluation.

**Active perception.**    In biological agents, perception is not a passive process but an active form of learning. Primates reduce uncertainty about their environment by selectively sampling sensory inputs that are most informative for their goals, a core idea in active learning theory [Findlay and Gilchrist, 2003, Gottlieb and Oudeyer, 2018]. Rather than processing the world holistically, agents continuously decide what to observe next, shaping how they learn and what they learn from.

In primate embodied vision, this principle is realized through foveation and saccadic eye movements [Hayhoe and Ballard, 2005, Yarbus, 2013]. Because high-acuity vision is confined to a small region of the retina (the fovea), primates construct a coherent understanding of their surroundings by actively moving their eyes to sample the world. Each fixation captures a small, high-resolution snapshot of the scene, while rapid saccades shift gaze to new locations. Over time, the visual system integrates these successive glimpses into a coherent representation [Melcher and Colby, 2008, Friston et al., 2012]. This process effectively turns perception into a form of sequential inference, where each movement reflects a prediction about which observation will be most useful next.

Multi-FoE captures this principle by restricting agents to foveated, egocentric observations and short-range saccade actions. Success therefore depends on learning how to look: selecting informative viewpoints, integrating partial observations over time, and constructing coherent internal world models from sparse, sequential glimpses. This tests active perception capabilities that conventional pixel-based RL benchmarks ignore.

**Boundary-free task switching and context inference.**    Real-world tasks rarely announce when one task ends and another begins; instead, contextual changes must be inferred from subtle shifts in sensory observations. For example, while foraging for food in a cluttered scene, a primate can abruptly switch goals if a predator appears or a conspecific competes for the same resource. In such moments, survival depends on the capacity to detect context transitions on the fly and adapt behavior without explicit external signals.

Multi-FoE reproduces this boundary-free multi-task setting by randomly interleaving tasks without any explicit signal to the agent. To perform well, an agent must infer the current task from its interactions alone, maintain task-specific knowledge over episodes, and flexibly adapt its strategy.

**Unknown goal states.**    Biological agents frequently pursue goals whose exact appearance is unknown. For example, when humans search for "the keys" or "something edible," they rely on abstract descriptions and must actively test hypotheses about what might constitute the target. This type of

goal-directed search where the agent is uncertain about what success looks like, is widespread in natural behavior but largely absent from standard RL tasks.

Multi-FoE operationalizes similar principle. Each task defines a hidden target zone, but the agent is never given a direct template of the target state. Instead, it must efficiently explore under strict interaction budgets to discover the target and obtain reward. The problem therefore becomes not simply where to search, but what to search for.

**Sample-efficient learning.** Biological agents learn under strict interaction limits. Humans and animals rarely experience identical situations repeatedly; instead, they must generalize from sparse, one-shot, or few-shot exposures. Children, for example, can learn new visual categories from just a handful of examples [Lake et al., 2015], and primates rapidly adapt their gaze strategies quickly when goals change [Hayhoe and Ballard, 2005]. In contrast, artificial agents typically require millions of interaction steps and extensive repeated exposure to the same environments.

Multi-FoE enforces sample efficiency by imposing strict limits on both the number of steps per episode, thereby constraining the agent's exposure to each task.

## B.2 Benchmark Formalization

We formalize our benchmark as a partially observable Markov decision process (POMDP).

**Observations.** At time step $t$, the agent occupies a location $\ell_t$ on a discretized $33 \times 22$ grid over a natural image. The observation function returns a local $84 \times 84$ crop centered at $\ell_t$:

$$o_t = \mathcal{O}(s_t, \ell_t),$$

where $s_t$ is the underlying image state. This enforces egocentric partial observability, as the agent never receives the full image in a single step.

The discretization of the image was defined such that each grid cell precisely corresponds to the target zone size employed in the human experiment. The images used and their corresponding target zones are shown in Figure 2.

**Actions.** The action space consists of short-range egocentric movements defined on a $9 \times 9$ neighborhood centered on the current location:

$$a_t \in \mathcal{A} = \{(-4, -4), \ldots, (0, 0), \ldots, (4, 4)\},$$

where each action shifts the agent's gaze within the image grid. This mirrors saccadic eye movements in primates.

**Rewards.** Each task $\tau$ defines a hidden target zone $z_\tau$ corresponding to a single grid cell. A reward is given if the agent's movement from location $\ell_t$ to $\ell_{t+1}$ passes through the target zone:

$$r_{t+1} = \begin{cases} +1, & \text{if } \mathrm{Line}(\ell_t, \ell_{t+1}) \cap z_\tau \neq \emptyset, \\ 0, & \text{otherwise.} \end{cases}$$

That is, the agent is rewarded not only when its new location lands inside the target zone, but also if the straight-line trajectory of its action intersects it. Episodes terminate immediately upon receiving a reward.

**Transition dynamics.** Given $s_t = (\tau, \ell_t)$ and $a_t = (\Delta x, \Delta y) \in \mathcal{A}$, the next location is $\ell_{t+1} = \Pi_{\mathcal{G}}(\ell_t + a_t)$, where $\Pi_{\mathcal{G}}$ clips to the grid boundaries. The task $\tau$ remains fixed within an episode and changes only between episodes according to the interleaving policy described below. Thus $\mathcal{P}(s_{t+1} \mid s_t, a_t)$ is deterministic over $\ell_{t+1}$ and constant in $\tau$ within an episode.

**Boundary-free task interleaving and step budget** During training, episodes are randomly sampled from a pool of 20 tasks with no boundary cues or signals indicating when the active task has changed. The agent must infer the current task context entirely from ongoing interaction. Each training episodes are limited to 25 environment steps. This corresponds to roughly 5 seconds of visual search time [Najemnik and Geisler, 2008].

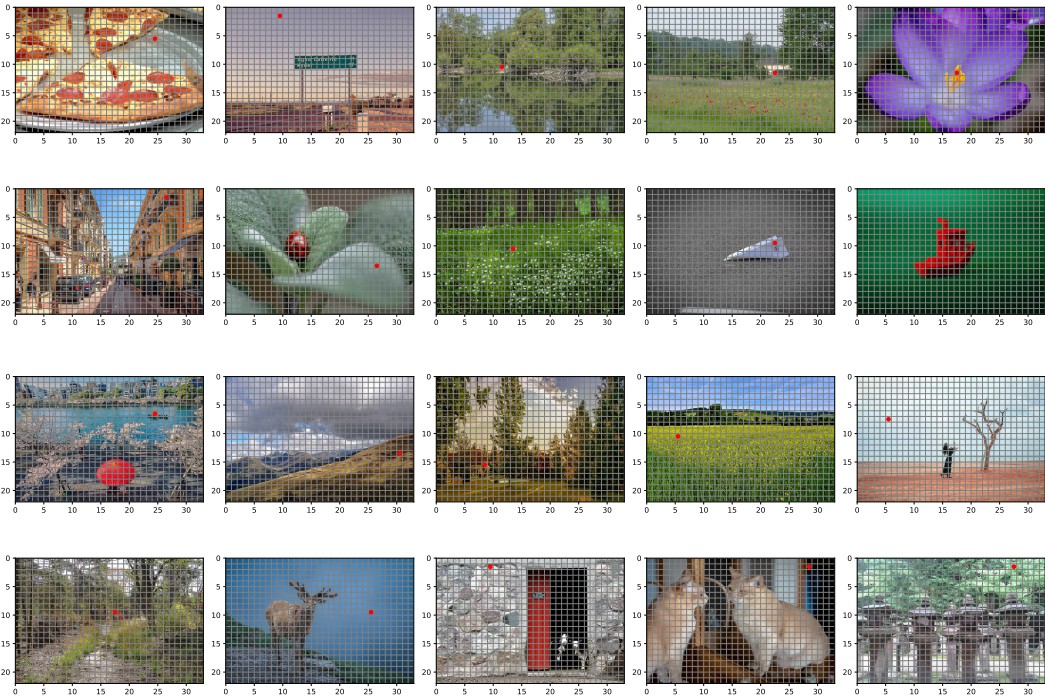

**Figure 2:** All of the task context scenes used in the benchmark and their corresponding hidden target zones (red dots).

