# OpenReview forum: "Context-Aware World Models for Task-Agnostic Control"
_NeurIPS.cc/2025/Workshop/UniReps — UniReps2025_

### Official Review · Reviewer_6ojL · 2025-09-15
**Strong Design, Valuable Benchmark, Narrow Baselines**

**Confidence:** 4

**Review:**

The paper introduces Hierarchical Memory World Models (HMWM), which split memory into environment-level (long-term) and task-level (working) components to capture global regularities while adapting quickly to specific tasks. Through self-supervised objectives and intrinsic rewards, the method enables fast in-episode learning and efficient exploration. On the new Multi-FoE benchmark, HMWM significantly outperforms strong baselines and approaches human-level efficiency under tight budgets.

Strengthens:

1. Clear and timely motivation for hierarchical memory in multi-task settings.

2. Crisp separation of EWM and TWM, with EWM conditioning task-specific predictions.

3. Well-designed Multi-FoE benchmark combining egocentric partial observability, interleaved tasks, and strict step limits.

4. Strong efficiency gains over BYOL-Explore while maintaining high reward.

5. Useful ablations and detailed implementation notes that aid reproducibility.

Weaknesses:

1. Heavy reliance on pretraining; performance depends strongly on seq-JEPA initialization.

2. Baselines are limited; comparisons to planning-based or meta-learning methods would make the case stronger.

**Score:**

4

**Topic Fit:**

3

---

### Official Review · Reviewer_FizH · 2025-09-16
**Promising hierarchical memory approach, unclear about relevance**

**Confidence:** 2

**Review:**

This work presents a new learning framework called Hierarchical Memory World Models, which integrates a long-term memory Environment-level World Model (EWM) with a short-term memory Task-level World Model (TWM). The EWM provides task-independent contextual knowledge, while the TWM focuses on learning task-specific dynamics. The paper also introduces Multi-FOE, a novel benchmark designed to evaluate model performance. Preliminary results suggest that the proposed hierarchical memory-based world models outperform several baselines.

Overall, the paper presents an interesting and promising idea: leveraging structured hierarchical models with both long-term and short-term memory. The contributions are sufficiently novel and developed for the extended abstract track.

My main question, however, is the relevance of the work to the UniReps workshop. While the methodological contributions are clear, the central focus appears to be on proposing a new learning approach rather than topics that directly align with the workshop’s call for papers.

**Score:**

3

**Topic Fit:**

1

---

### Official Review · Reviewer_a78s · 2025-09-16
**Memory model for primate vision like tasks**

**Confidence:** 3

**Review:**

The paper introduces Hierarchical Memory World Models (HMWM), a framework that separates environment-level world models (long-term memory) and task-level world models (working memory) to improve multi-task adaptation. A new benchmark, Multi-FoE, is proposed to test agents under biologically inspired constraints such as partial egocentric observations, unknown goal states, and boundary-free task switching. Experiments show that HMWM outperforms baselines such as BYOL-Explore and approaches human-level efficiency in low-budget regimes.

Strengths:
1. Aligning hierarchical memory structures with world models is original and well-motivated by neuroscience.
2. Multi-FoE offers a realistic and challenging testbed, inspired by primate vision and human learning studies.
3. Includes comparisons with oracle and random baselines, human baselines, and ablation studies.
4. Demonstrates clear gains in sample efficiency and adaptation speed, particularly with pretrained backbones.

Weakness:
1. Multi-FoE is interesting but limited to visual search; unclear how results transfer to broader multi-task RL domains (e.g., control, language).
2. Without pretrained backbones, improvements are modest; much of the gain comes from initialization rather than architecture.

**Score:**

4

**Topic Fit:**

2